# Use of Digital Educational Technologies among Nursing Students and Teachers: An Exploratory Study

**DOI:** 10.3390/jpm11101010

**Published:** 2021-10-08

**Authors:** Fernanda Loureiro, Luís Sousa, Vanessa Antunes

**Affiliations:** 1Centro de Investigação Interdisciplinar Egas Moniz (CiiEM), Escola Superior de Saúde Egas Moniz, 2829-551 Almada, Portugal; vanessa2em@gmail.com; 2Comprehensive Health Research Centre (CHRC), Nursing Department, Universidade de Évora, 7004-516 Évora, Portugal; lmms@uevora.pt

**Keywords:** education, distance, education, learning, digital

## Abstract

The emergence of digital educational technologies (DET) raises questions regarding the personalization of both teaching and care. DET use implies profound changes with consequences in nursing care and in nursing teaching-learning process. With the purpose of contributing to the improvement of the teaching-learning process through the use of DET, an exploratory-descriptive, cross-sectional, and observational study, with a quantitative approach (descriptive and inferential statistics), was developed. Online questionnaires were applied (*n* = 140 students and *n* = 23 teachers) after ethics committee approval. Results point to low cost and access without time/space limits as the main benefits, and decreased interaction, less physical contact, and technical difficulties as constraints. Globally, there was no difference between students and teachers in the use of DET. Still, men report more constraints than women. In this sample, the use of DET is still at an early stage. Both students and teachers are still unfamiliar with the scope and possibilities of these tools, not taking full advantage of the potential they have to offer. The impact of DET used in personalized nursing care is still yet to be understood.

## 1. Introduction

Digital educational technologies (DET) have been used increasingly in recent years due to technological evolution. It implies profound changes in teaching practices with consequences in the teaching-learning process [1]. The use of platforms not only for providing academic content, but also for distance learning, have a great impact on both teachers and students: in the format of the contents that are made available, in the language used, class duration, time management, methods and hours of study, and in communication strategies. The recent public health pandemic emphasized the need for digital technologies use to maintain activity with social distance. In this context, DET represents an added value since they increase the interactivity and promote additional space for knowledge construction [2]. DET can be used in an asynchronous or a synchronous way. When used in its asynchronous form, it offers as added benefits: the possibility of a flexible schedule, the availability of quality material at low cost, and the potential to reach a high number of students without geographical limits [3]. However, issues on humanization of education and knowledge systematization are raised and, additionally, although DETs offer a wide range of possibilities, they do not guarantee learning quality [4]. Still, there is some reluctance in its use, particularly by teachers, as DETs break with traditional education, placing the emphasis on the student as the protagonist of the teaching-learning process [5].

As technology evolves, it has been integrated into different areas where education is included. Having this in mind, DET can be defined as the use of technological resources to improve teaching and learning, promote educational development and access to data and material [1]. It implies the use of virtual scenarios, online platforms, and digital resources, among others. Historically, educational sciences have used DET as a resource to improve teaching since the 80s, but in nursing its use is more recent. Benefits can be pointed out from its use, such as improved interaction and content integration [2,4], easy access, more dynamism and interaction with students outside a physical school space [1]. However, constrains can also be pointed to, namely financial (adopting state-of-the-art technology can be expensive) and personal (teachers may be reticent to change traditional educational methods). It is not integrated in any theory of learning, but it fits a diverse theoretical framework. A major element of this theoretical frameworks is socio-constructivism that underlines the role of social processes in individuals’ learning [6]. Social constructivism-based theory understands learning as a process that occurs through supportive collaboration with other people and leads to knowledge and evolution [7].

In nursing, the use of DET is seen as a resource that is complementary to traditional teaching [4] with increasing use and good results such as enhanced nursing students’ problem-solving skills [6]. An example of the use of DET with other educational strategies is its use combined with case studies improving its benefits [8]. However, Chavaglia et al. [9] pointed to the need for greater diversification in the use of DET. In their study with undergraduate nursing students, they concluded that digital power point presentations, email and a google search engine were the predominant digital tools used by students. 

A link between the use of DET, nursing education and sustainability is also found in the literature. Education in sustainable development is recognized as important particularly in health professions such as nursing. The use of DET offers an opportunity to reinforce this link since DET are sustainable and can be applied with low environmental impact [10]. Other innovative andragogical approaches are also being used, such as augmented reality and virtual simulation. Foronda et al. state that results from the use of these technologies are encouraging, since they suggest efficacy in improving nursing students learning outcomes [11].

However, some caution is required in the implementation of DET, especially by teachers. They have to be conscious of factors that influence the success in DET’s application, and a minimum quality requirement has to be assured [10]. Usually, the types of DETs used by teachers are limited to those available in their institution and with easy access [12], which limits the quality of teaching and consequently the quality of the learning process. On the other hand, there has been a greater trend towards personalization of care and patient-centeredness. The use of this type of technology in teaching and its impact on care is still unknown. However, it seems to be consensual that DET may expand nursing education and consequently improve patient outcomes [11].

For the present study non-presential classroom was considered as classes that occur without physical presence in the same room of both students and teachers. The purpose of this study is to contribute to the improvement of the teaching-learning process in the nursing undergraduate degree through the use of DET in non-presential classroom teaching. The objective is to evaluate the use of DET in non-presential classroom teaching, benefits, constraints, and implications for teaching-learning process in nursing students. The following questions were defined: how is the use DET fulfilled in non-presential classroom teaching in nursing? What are the benefits and constrains to the use of DET? What are the implications for teaching-learning process of nursing students with the use of DET?

## 2. Materials and Methods

### 2.1. Study Design

An exploratory-descriptive, cross-sectional, and observational study was outlined. The research design fits into the quantitative paradigm with the use of surveys as a data collection technique. 

The study was approved by the school board as well as by the school ethics committee. It was an independent study not integrated in any course or curricular unit. The study was applied to 230 inquiries and invitation to participate in the study was sent by email with a survey link attached. Prior to fill the survey information regarding framework, aim, confidentiality, ethical issues and researchers contacts was provided. Consent was obtained prior to data collection. Participants had to mandatorily give their consent to advance with participation. 

### 2.2. Sample and Setting

The population consists of all nursing students and teachers who lecture the undergraduate nursing course at a private nursing school in Lisbon, Portugal. The sample is therefore of simple non-probabilistic type. Students from all academic years were invited to participate (*n* = 205). Teachers who teach classes of any type (*n* = 25) were selected and teachers involved solely in clinical nursing were excluded.

### 2.3. Instrument

The online survey designed for this study includes 60 items based on previously published literature [13,14,15,16]. It had two parts: first part with sociodemographic data such as sex, age, and type of experience in the use of DET. Additional data collected included academic year for students’ and professional experience and type of employment for teachers. The second part of the survey had questions related to the use of DET, selected and adapted from a previous literature review. 

Types of DET [13] included it a 5-point Likert scale with 23 items within 5 dimensions (organizational tools, communications tools, presentation tools, learning assessment tools and identity transformation tools). Benefits of using DET include 10 dichotomous items and were retrieved from Kokol et al. [14]. These authors applied a survey to 125 nursing students’ study and identified benefits and challenges of online education. Constraints of using DET was adapted from Lloyd et al. [15] and comprise 13 dichotomous items. The original scale included 22 items and authors performed exploratory factor analysis for validity and Cronbach alpha for reliability. Four factors were extracted that had high reliabilities (0.892; 0.806; 0.805; 0.870). In our study, we adapted the items for our context. Finally, implications in the use of DET include 14 dichotomous items adapted from an integrative review on digital technologies in the teaching of nursing skills by Silveira and Cogo [16]. 

Before its application, the surveys were sent to 5 experts. Content validity index was calculated as described by Polit and Becket [17] to verify the extend of expert agreement. A value of 0.90 was obtained which is considered adequate by the same authors. 

Additionally, for instrument validity, to understand how types of DET were related and to favor statistical analysis, exploratory principal components factor analysis with varimax rotation was performed on the 23 items that addressed types of DET. The Kaiser-Meyer-Olkin measure verified the sampling adequacy for the analysis, KMO = 0.759. Additionally, Bartlett’s sphericity test indicated that correlations between items were sufficiently large, X^2^ = 1146.772, *p* < 0.001.

Initially, for the first-dimension sevens factors were identified that explained 64.9% of variance. However, two item (calendars and electronic portfolio) had correlations <0.3. By removing them and forcing analysis to the initial 5 factors that were identified in the literature, a 53.8% of the variance was obtained as explained on Table 1. Regarding communalities values ≥0.4 were considered. We also included the item: “Instant messages” although h^2^ value is 0.386 because it is a marginal value and due to its conceptual importance in the context of this study.

For reliability assessment the Cronbach Alpha coefficient as well as the Kuder-Richardson formula were determined. Regarding types of DET, the alpha Cronbach values range between 0.594 in factor 5 and 0.739 in factor 1 as shown in Table 1. For benefits, constraints and implications thar are dichotomous items in the Kuder-Richardson formula indicated the following values: 0.670 (benefits); 0.502 (constrains) and 0.584 (implications). The values obtained are considered satisfactory [18], and additionally, it must be taken into consideration that it is a first attempt to use this instrument.

### 2.4. Procedure

Participant’s recruitment was performed by email. Since this was an independent study, not integrated in any course or discipline, researchers contacted students and professors through email addresses. An email was sent to all the participants inviting them to participate in the study. This email contained a survey link that allowed access to the online instrument. It was available for 1 week, and after that time, a reminder was sent, and the survey was available for one week more. The number of answered surveys were monitored by the researchers at the end of each week. After two weeks had passed the link was removed and data was transferred for analysis. All surveys were fully completed by participants.

### 2.5. Data Analysis

Descriptive and inferential statistics were used for data analysis. Statistical analyses was performed with SPSS statistical package, 27.0 version (SPSS, Inc., Chicago, IL, USA). Categorical variables are presented as percentages. All variables followed a non-normal distribution so nonparametric test namely Mann-Whitney U test was used for comparisons between groups. To verify association between categorical variables Phi/Cramer V was used. A *p* value of 0.05 or less was considered statistically significant.

## 3. Results

### 3.1. Sample Characteristics

Of the study population, 163 subjects (140 students and 23 teachers) completed and returned the surveys, so the response rate was 70.8%. Most respondents were female (85%; *n* = 138) and had no training or formal education on DET, only experience as users (86%; *n* = 140). Sociodemographic characteristics are summarized in Table 2. 

### 3.2. Types of DET 

Regarding the type of DET used, they were grouped into the five categories identified in factorial analysis. For presentation data, answers were clustered into two group: very frequently and frequently answers and occasionally, rarely or never answers. Results are summarized in Table 3.

Mann-Whitney U test was used to assess if there were association between student’s sample and teachers sample and the use of DET. No significative association was found except on the item “tests” (U = 2094.5; *p* = 0.016).

As to technologies used, results show the use of the technologies provided and available in the institution where the survey was applied: Moodle (98.8%; *n* = 161) and Microsoft Teams (98.2%; *n* = 160). Other technologies identified included Zoom (22.1%; *n* = 36), Mentimeter (4.9%; *n* = 8), Kahoot (4.3%; *n* = 7), Skype (2.5%; *n* = 4) and Google classrooms (1.2%; *n* = 2).

As to type of classes DET they were mostly used in theoretical class (93%; *n* = 153). In practical and laboratory classes DET were used less frequently (37% and 28%, respectively).

### 3.3. Benefits and Constrains in the Use of DET

Regarding benefits and constraints responses are summarized in Table 4.

To assess association between type of inquiries (students or teachers) Phi/Cramer’s V was calculated. There were only significant statistical differences between teachers and students in two items: “it allows the integrating of multiple learning tools (pdf, links, app, among others)” and “promotes student accountability”. Teachers refer to more benefits in the integration of multiple learning tools (V = 0.170; *p* = 0.030), and students attribute more benefits in item: “promotes accountability” (V = 0.170; *p* = 0.030). Regarding constrains, the same procedure was followed; however, no significant statistical differences were found.

### 3.4. Implications in the Use of DET

The last question of the questionnaire referred to implications in the use of DET. Answers are synthetized in Table 5.

Regarding implications also Phi/Cramer’s V was calculated to assess association between type of inquiries (students or teachers) and implications. Nevertheless, there were no significant statistical differences between the groups. 

Globally, there was no difference between students and teachers in the use of DET. Still, when analysis is performed by gender differences were found regarding Factor 3 in types of DET and constrains.

Women got a higher score concerning the types of DET used (Factor 3) compared to men, that is, women use these resources more than men (U = 1531.5; *p* = 0.040). Still, men report more constraints than women in the use of DET (U = 2255; *p* = 0.005). 

As stated earlier, all questions had an open space for additional answers; however, there were few answers and those who did respond mentioned aspects already identified.

## 4. Discussion

This study allowed us to gather evidence related to the use of DET, being a first step to the improvement of the teaching-learning processes in the nursing undergraduate degree. Our results show a scarce use of DET by both teachers and students. Literature points to an increase in the use of DET in nursing teaching [16]. Our results show that there is space for improvement in this area. A possible explanation for this result may be the existence of interpersonal and institutional barriers, training and technological constrains and the lack of cost/benefit analysis [15].

Overall, 86% of our sample had no formal training on DET, which is in line with the literature. However, studies highlight that digital technologies are used both with academic and personal purposes [19]. Probably teachers in this sample do not feel the need for formal education as the day-to-day use of these tools allows them to use them without limitations or constraints. Additionally, digital tools are, naturally intuitive and user friendly. When considering our sample of students, they belong to a generation raised with information technology, internet, and social networks, which make them more confident manipulating new platforms and devices, not really requiring training in the area.

As mentioned in the results chapter, the statistical procedure allowed to group the types of DET in five new factors, categorized as follows: Training and Discussion Tools (factor 1); Communication tools (factor 2); Presentation and assessment tools (factor 3); Organization tools (factor 4); and Complementary learning tools (factor 5).

The training and discussion tools (factor 1) included the following items: discussion forums, blogs, avatars, virtual scenarios, and immersive technology. In this category, results show that avatars and virtual scenarios are poorly used. This types of DET are identified as particularly relevant in nursing education since they may provide a solution to issues such as: faculty shortages, scarcity of clinical placements and limited onsite laboratory space [11]. Additionally, its use allows students to repeatedly simulate procedures and care or even recreate high risk events that they may not contact within clinical context [11,16].

Communication tools category (factor 2) includes: videocalls, web conferences, audio, and video. Videocalls (74%) and audio (74%) are the preferred communication tools for the total sample. Although web conferences can save students time and prevent the inconveniences of traveling, video and audio conferencing provide the benefit of visual aids, allowing participants to make use of multiple senses, improving their concentration levels and increasing their capacity to absorb more information [20,21].

Within the presentation and assessment tools category (factor 3), the following items were identified: slides, tutorials, image sharing, bibliography sharing, activities, and tests. Tutorials got the lowest score (39%). This technology is still underused in our sample, but perhaps the changes in the lifestyle of today’s society could lead to a greater need and recognition of its use in this setting. The traditional classroom can restrict daily life, as it requires anticipated planning. Otherwise, the tutorial can be paused, rewound or fast-forwarded, and the lecture can be heard as many times as needed. These characteristics fit the current generation of students that crave the digital world and are extreme consumers of technology [22]. On the other hand, slides got the most representative score, as it is used by 89% of our sample. The use of DET to share slides can be framed as a simplistic and traditional use of this type of technologies. 

In this category, it is also possible to verify an association between gender and the use of presentation and assessment tools, as women got a higher score compared to men (U = 1531.5; *p* = 0.040). Additionally, men report more constraints than women in the use of DET (U = 2255; *p* = 0.005). These results are quite interesting, as there is the general idea that women are outnumbered in informatics. Nevertheless, literature shows that the very first computer programmers and IT users were women, and that technology are more frequently used in female-dominated areas [23], such as teaching and nursing. Cai et al. [24] found that there are no significant gender differences in the attitudes toward technology. However, these authors argue that in the academic context, women are more prepared for technology use than the general female population.

In types of DET, factor 4 covers the following organization tools: online schedule, mind maps or graphic organizers, and file and management store. This last one is the most used by 67% of our sample. It is a more cost-effective system, as it reduces organization bureaucracy and saves space. Students can easily search, access, and share files. Additionally, support for decision-making and knowledge discovery can be achieved in an effectively way through the use of massive amounts of data that can be easily stored [24].

Instant messages and chats, research, and social network were considered as complementary learning tools (factor 5). Instant messages and chats (84%) are the mostly used by both students and teachers in our sample. They may be more relevant because they are one of the oldest communication tools, and usually a quick option when network failures occur. Within this category, the results also demonstrate that DET are used for research by 72% of the sample. These results are aligned with literature that shows that traditional libraries are being partially replaced by digital search, specially by students who prefer it because they are easy and quick to search [25]. 

More than half of our sample identified the following benefits of using *DET*: the integration of multiple learning tools (66%), low cost (68%) and access without time or space limits (71%). Pinto and Leite [19] also mention that the use of DET has effects on the interaction time between students and teachers extending it beyond the traditional academic period. This access 24/7 can also be seen as a constraint and reduce teacher’s quality of life. However, considering the students’ perspective, it can be a major benefit. Männistö et al. [6] reported that the use of DET in nursing teaching enhances motivation for learning. When used in nursing education DET can improve the learning experience particularly in clinical learning settings [26]. Regarding low cost, there is no robust evidence that e-learning is a more cost-effective way to deliver knowledge when compared to traditional methods [27]. Yet, for students and teachers it may prove to be a more economical option as it avoids travel or food expenses.

Still regarding the benefits, no significant associations were identified between the teacher and student groups, except for the items: it allows the integrating of multiple learning tools (pdf, links, app, among others)” and “promotes student accountability”. These results are quite surprising because, given the age difference between the two groups in our sample, a more pronounced association would be expected. Literature shows that older adults are less likely to use technology that younger adults [28,29]. 

Our results show that constraints had greater statistical evidence than benefits. The more relevant were: the decreased interaction between students/teachers (72%), the less physical contact (77%), and the technical difficulties (67%). These aspects are pointed as the main barriers to the use of DET in literature and in fact, digital technologies can be used to enhance education, but they cannot completely replace face-to-face teaching [26]. Regarding technical difficulties, Naveed et al. [30] showed that efficient technology, infrastructure readiness and system reliability are some of the main critical success factors in implementing DET. To overcome these constrains several measures can be taken such as the use of combined methodologies (both presential and non-presential), a balanced and flexible schedule that maintains physical contact between students and teachers and permanent technical support available for users.

Implications identified by more than half of our sample include both the possibility of using software in simulation scenarios (57%) and the possibility to repeat simulations until learning, ensuring patient safety (55%). Overall, 66.9% of our sample consider that DET stimulates students’ independence (66%) and stimulate self-learning (71%). In nursing, the possibility to repeat simulations is identified as important and relevant for learning [31]; additionally, it is a significant factor for patient safety. Nevertheless, in our sample, only 14% considered that the use of DET improves performance in the execution of techniques. This could be due to a broader interpretation of the DET concept, as the technology by itself does not improve the practical skills required for nurses. Rather, the use of specific technology tools, such as clinical virtual simulation, can be used as a complementary strategy to DET, improving clinical reasoning skills [31]. 

The impact of DET on nursing care and patient outcomes is still poorly explored. Although there seems to be a link between improved nursing education and improved patient outcomes, this issue must be further explored. Additionally, there is a trend towards more centered and personalized models of care and the use of massive forms of education may not be suitable to this trend. It should be noted that some of the digital tools used in education are used in the health area, also with the aim of personalizing and, in this case, improving care. With regard to students, DET may be adapted to their educational and personal needs, which translates into a more student-centered teaching. Additionally, due to its positive impact, these technologies are being implemented in continuing education and professional development. A good example is the use of immersive technologies that allow nurses to have dynamic experiences with patients and be more prepared for real-world clinical settings. Nevertheless, the impact of DET on nursing care and patient outcomes is still poorly explored. Although, there seems to be a link between improved nursing education and improved patient outcomes, this issue should be further studied. It is also worth exploring if more student-centered teaching is later reflected in more people-centered nursing care.

Palvia et al. [32] state that e-education is advancing and is here to stay all around the world, which brings implications such as the need for improvement of telecommunications infrastructure, the acknowledgement of online education as equivalent to traditional face to face education and the globalization of e-education, which is inevitable, similar to the globalization of email or e-commerce. This authors also state that both online (virtual) and offline education must be combined so the virtues of both can be used.

It is necessary to have a more in-depth knowledge on this subject and a greater investment, both by teachers and educational institutions. Additionally, to understand the impact of the use of these technologies on nursing students learning, both in academic and clinical contexts, is needed. Nursing schools must invest in updated information technologies appropriate to the nursing curriculum, as well as providing training on their use to teachers and students. In our study, we verify that both teachers and students use mainly the tools made available by the school. Therefore, one way to improve the use of DET is to make more tools available. Additionally, in our setting the current teacher’s generation is more familiar with the first tools that were created and used as DET so those are the ones they mostly use. In our sample, most teachers do not have formal education on the use of DET and this should be included in the annual training program.

As to limitations this study was applied after the first lockdown that occurred in Portugal which triggered an exponential growth in the use of DET. As more periods of lockdown occurred both teachers and students were forced to improve the use of DET. Therefore, we consider that if the application of this study had occurred after these periods the results would have been different. Study design and type of sample are also limitations since our exploratory-descriptive study used a convenience sample and therefore results cannot be generalized.

## 5. Conclusions

Our results show that the use of DET in this sample is it is still at an early stage. Both students and teachers are still unfamiliar with the scope and possibilities of these tools, not taking full advantage of the potential they have to offer. The integration of multiple learning tools, low cost, and access without time or space limits are considered the main benefits of DET. On the other hand, the decreased interaction between students/teachers, the less physical contact, and the technical difficulties, are seen as the greater constraints. The main implications of DET are the possibility of using software in simulation, and the possibility to repeat simulations until learning, ensuring patient safety. It is also considered as a method that stimulates students’ independence and self-learning.

Although distance education through DET is a few years old, as far as nursing education is concerned, this concept is still very recent. Globalization and, more recently, the pandemic context that forced social isolation, have further boosted the introduction of DET in nursing education. Distance education thus becomes an effective strategy to reach people who want or need to be qualified, but who, for different reasons, cannot depart from their context of life and work [2]. Nevertheless, this constitutes a challenge to the nursing traditional teaching-learning methods, which have a predominantly practical and proximity component [27]. The distance between students and teachers, should be used to its full potential, involving students dynamically in the learning process, respecting independence, and autonomy, establishing links between learning and life and professional experience. On the one hand, it is necessary to provide teachers with skills to establish a link with students and stimulate their learning and engagement. It is also recommended that schools select the appropriate DET methods for teaching nursing, namely investing in robust digital platforms and state-of-the-art simulated practice technology [32]. The link between improved nursing education and its implication on nursing care must also be further explored.

## Figures and Tables

**Table 1 jpm-11-01010-t001:** Exploratory factorial analysis for types of DET.

Types of DET	Factors					h^2 *^
	1	2	3	4	5	
Discussion Forums	0.466					0.533
Blogs	0.653					0.490
Avatars	0.715					0.606
Virtual Scenarios	0.709					0.599
Immersive Technology	0.744					0.649
Videocalls		0.580				0.470
Web conference		0.429				0.415
Audio		0.762				0.655
Video		0.763				0.645
Slides			0.715			0.587
Tutorials			0.481			0.526
Image sharing			0.593			0.556
Bibliography sharing			0.653			0.597
Activities			0.540			0.590
Tests			0.611			0.613
Online Schedule				0.713		0.545
Mind maps or graphic organizers				0.703		0.579
File and management store				0.531		0.515
Instant messages					0.543	0.386
Research					0.697	0.589
Social network					0.730	0.665
Eigenvalues	2.80	2.47	2.38	2.11	2.04	
% of variance	13.35	11.77	11.34	10.06	9.72	
Cronhbach Alpha	0.739	0.667	0.715	0.602	0.594	

*-h^2^: communalities.

**Table 2 jpm-11-01010-t002:** Sample sociodemographic characteristics.

Variables	*n*	%
Sex		
Female	138	85
Male	25	15
Age (students)		
<20 years	48	28
21–30 years	84	52
>31 years	8	5
Age (teachers)		
<30 years	2	9
31–50 years	5	22
>51 years	14	61
Curricular Year (students)		
1st	35	25
2nd	33	24
3rd	56	40
4th	16	11
Teaching experience (teachers)		
<5 years	5	22
6–10 years	2	9
11–20 years	10	43
>21 years	6	26
Experience in using DET		
Experience as user	140	86
Did or is doing a course	21	13

**Table 3 jpm-11-01010-t003:** Results regarding type of DET used by teachers and students.

	Total Sample (*n* = 163)				Students (*n* = 140)				Teachers (*n* = 23)			
Types of DET	VF and F		O, R or N		VF and F		O, R or N		VF and F		O, R or N	
	*n*	%	*n*	%	*n*	%	*n*	%	*n*	%	*n*	%
**Factor 1**												
Discussion forums	22	13	141	87	14	10	126	90	8	35	15	65
Bloggs	18	11	145	89	16	11	124	89	2	9	21	91
Avatars	13	8	150	92	11	8	129	92	2	9	21	91
Virtual scenarios	12	7	151	93	10	7	130	93	2	9	21	91
Immersive technology	22	13	141	87	20	14	120	86	2	9	21	91
**Factor 2**												
Videocalls	121	74	42	26	104	74	36	26	17	74	6	26
Web conferences	78	48	85	52	65	46	75	54	13	57	14	43
Audio	121	74	42	26	107	76	33	24	14	61	9	39
Video	113	69	50	31	100	71	40	29	13	57	10	43
**Factor 3**												
Slides	145	89	18	11	123	88	17	12	22	96	1	4
Tutorials	63	39	100	61	52	37	88	63	11	48	12	52
Image sharing	120	74	43	26	102	73	38	27	18	78	5	22
Bibliography sharing	107	66	56	34	87	62	53	38	20	87	3	13
Activities	109	67	54	33	91	65	49	35	18	78	5	22
Tests *	109	67	54	33	**96**	**69**	**44**	**31**	**13**	**57**	**10**	**43**
**Factor 4**												
On-line schedule	42	26	121	74	30	21	110	79	12	52	11	48
Mind maps or graphic organizers	46	28	117	72	41	29	99	71	5	22	18	78
File and management store	110	67	53	33	92	66	48	34	18	78	5	22
**Factor 5**												
Instant messages and chats	137	84	26	16	119	85	21	15	18	78	5	22
Research	118	72	45	28	108	77	32	23	10	43	13	57
Social network	98	60	65	40	95	68	45	32	3	9	20	91

VF—very frequently; F—frequently; O—occasionally; R—rarely; N—never; * significant at *p* < 0.05.

**Table 4 jpm-11-01010-t004:** Results regarding benefits and constraints in the use of DET.

	Total Sample (*n* = 163)		Students (*n* = 140)		Teachers (*n* = 23)	
**Benefits of using DET**	* **n** *	**%**	* **n** *	**%**	* **n** *	**%**
It allows access to educational content without time/space limit;	115	71	51	37	15	65
Low cost;	110	68	21	15	4	17
It allows the integrating of multiple learning tools (pdf, links, app, among others); *	108	66	35	25	9	39
Promotes student accountability; *	96	59	50	36	4	17
Improves the quality of the teaching-learning process;	67	41	33	24	9	39
Enhances multidisciplinary work;	54	33	101	72	14	61
Improves the student’s ability to analyze, synthesize and think critically;	45	28	38	27	7	30
Improves students’ preparation for applying theoretical content to practice;	44	27	95	68	15	65
Increases students’ creativity, motivation, and quality of care;	42	26	86	61	10	43
Improves the presentation of nursing care sensitive results.	25	15	92	66	16	70
**Constraints of using DET**	* **n** *	**%**	* **n** *	**%**	* **n** *	**%**
Less physical contact with students;	126	77	111	79	15	65
Decreased interaction between students/teachers;	117	72	73	52	13	57
Technical difficulties (network failure, server overload);	109	67	72	51	17	74
Difficulty in obtaining visual feedback;	89	55	99	71	18	78
Depersonalization of teaching;	86	53	11	8	10	43
Moving away from traditional teaching models;	85	52	43	31	10	43
Lack of preparation in the use of DET;	53	33	21	15	5	22
Difficulty in managing time and content;	43	26	99	71	10	43
Lack of infrastructure and technologies;	26	16	8	6	5	22
Lack of regulation in the use of DET;	21	13	39	28	4	17
Constant updating and innovation of DET;	13	8	79	56	6	26
Resistance to the use of DET;	8	5	6	4	2	9
Fear of technology.	7	4	6	4	1	4

* significant at *p* < 0.05.

**Table 5 jpm-11-01010-t005:** Results regarding implications in the use of DET.

	Total Sample (*n* = 163)		Students (*n* = 140)		Teachers (*n* = 23)	
Implications in the Use of DET	*n*	%	*n*	%	*n*	%
Implications—Simulations						
Possibility to repeat simulations until learning, ensuring patient safety;	90	55	78	56	14	61
Possibility of using software in simulation scenarios;	93	57	73	52	16	70
Promotion of meaningful learning;	51	31	30	21	6	26
Ensures better practical performance for the student.	36	22	53	38	11	48
Implications—Stimulation of learning						
Arouses students’ curiosity;	47	29	73	52	16	70
Stimulates students’ independence;	108	66	95	68	12	52
The existence of multiple tools makes learning more stimulating;	57	35	45	32	3	13
It allows to personalize the teaching;	48	29	33	24	14	61
Improved use of theoretical content.	49	30	48	34	9	39
Implications—Learning skills						
Stimulate self-learning;	115	71	29	21	8	35
It makes learning less monotonous;	39	24	37	26	6	26
Existence of a safe environment allowing errors that lead to improved technical execution;	51	31	18	13	4	17
Decreases anxiety when performing techniques;	44	27	102	73	13	57
Improves performance in the execution of techniques.	22	14	46	33	8	35

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
