# Peer review of "Use of Digital Educational Technologies among Nursing Students and Teachers: An Exploratory Study"

_jpm, 2021, doi:10.3390/jpm11101010_

Round 1
Reviewer 1 Report
Thank you for your attention to this meaningful and relevant topic in nursing education.
- Line 26: Recommend changing “in latest years” to “in recent years.”
- Line 28: When you say that DET “implies profound changes in teaching practices with consequences in the teaching-learning process,” can you explain more or give examples?
- Line 41: Much more explanation regarding DET and what it is is necessary, what is the history behind it, pros, cons, examples, etc.
- Line 47: consider replacing the word “evolution” with specific skills obtained with DET (ie. better problem solving, etc.)
- See below
Line 59: change pedagogy to andragogy. Differences Of Pedagogy Vs Andragogy In eLearning
The terms “andragogy” and “pedagogy” are of Greek origin, both including the Greek verb “ago”, which means “guide”, and the Greek words “andras” (man) and “paidi” (child) respectively. Thus:
Pedagogy = paidi (child) + ago (guide)
Andragogy = andras (man) + ago (guide)
You can conclude from the above that pedagogy is a child-focused teaching approach, whereas andragogy an adult-focused teaching approach; or, formally, pedagogy is the art and science of helping kids learn, whereas andragogy is the art and science of helping adults learn.
Reference: https://elearningindustry.com/pedagogy-vs-andragogy-in-elearning-can-you-tell-the-difference
- Line 60: clarify your sentence. What preliminary results are you referring to exactly?
- For paragraph starting line 70: define non-presential for the audience.
- Line 81: can your description of the study approach be simplified?
- Line 82: remove comma after the word “surveys”
- Line 83: the last sentence here should go into the data analysis section.
- Line 85: is the ethics committee the same as an IRB (Institutional Review Board)? If so, maybe give a short explanation. Also, specify that you obtained permission prior to any data collection.
- For the paragraph starting line 84: explain more about how students gave consent. Were they given the option to participate or not? Was this study part of a course, or something completely separate from a course. How did you prevent coercion?
- Paragraph starting line 104: this paragraph could be more clear and concise.
- Line 111: what was the cronbach’s alpha in the published study?
- For paragraph starting line 130: are these Cronbach alphas from the current study?
- For the paragraph starting line 139: more detail needs to be written that explains more about the recruitment and consent process, especially since students were used as subjects. Also, did you include only fully completed surveys? Or did you include partially completed ones as well?
- For line 153: I’m not clear on the results here; did the students and teachers complete the same surveys? If so, was it from the perspective of participation in DET? Not about usage of the faculty and participation for the students? This needs to be clarified.
- Line 207: I’m not convinced that the aim of the paper is to contribute to the improvement of educational processes….it seems the purpose is more likely aimed at gathering preliminary evidence related to the use of DET…
- For paragraph starting line 207: I feel like your explanation related to why faculty aren’t using DET more frequently should be backed with evidence from the published literature. Is this related to their age, generation, etc.?
- For paragraph starting 288: add more detail about how we can overcome constraints
- Line 329: need to include study design, convenience sample as limitations.
- There should be citations in the conclusion section.
Author Response
Response to Reviewers
We would like to thank the editor and the reviewers for careful and thorough reading of this manuscript and for the thoughtful comments and constructive suggestions, which were considered in the new manuscript version.
After receiving the reviewer’s comments, changes were made in the manuscript as detailed below:
Reviewer 1
1.Line 26: Recommend changing “in latest years” to “in recent years.”
Thank you form the suggestion. Changes were made according to recommendation.
2. Line 28: When you say that DET “implies profound changes in teaching practices with consequences in the teaching-learning process,” can you explain more or give examples?
We appreciate the comment. Additional text was added to explain this sentence.
3.Line 41: Much more explanation regarding DET and what it is necessary, what is the history behind it, pros, cons, examples, etc.
Thank you for the comment. We have revised the text to address your concerns and hope that it is now clearer.
4.Line 47: consider replacing the word “evolution” with specific skills obtained with DET (ie. better problem solving, etc.)
We appreciate the comment. In this particular phrase “Social constructivism‐based theory understands learning as a process that occurs through supportive collaboration with other people and leads to knowledge and evolution”, the word evolution refers to learning and not to DET, therefore we consider that the word “evolution” should be kept.
5. See below Line 59: change pedagogy to andragogy. Differences Of Pedagogy Vs Andragogy In eLearning The terms “andragogy” and “pedagogy” are of Greek origin, both including the Greek verb “ago”, which means “guide”, and the Greek words “andras” (man) and “paidi” (child) respectively. Thus: Pedagogy = paidi (child) + ago (guide) Andragogy = andras (man) + ago (guide) You can conclude from the above that pedagogy is a child-focused teaching approach, whereas andragogy an adult-focused teaching approach; or, formally, pedagogy is the art and science of helping kids learn, whereas andragogy is the art and science of helping adults learn. Reference: https://elearningindustry.com/pedagogy-vs-andragogy-in-elearning-can-you-tell-the-difference
Thank you for the clarification. Changes were made according to recommendation.
6. Line 60: clarify your sentence. What preliminary results are you referring to exactly?
Thank you for the comment. We have revised the text to address your concerns and hope that it is now clearer.
7. For paragraph starting line 70: define non-presential for the audience.
Thank you form the suggestion. Changes were made according to recommendation. Please see lines 84-85.
8. Line 81: can your description of the study approach be simplified?
We appreciate the comment. For study classification we used Burkett approach (https://pubmed.ncbi.nlm.nih.gov/2182361/) that considers three dimensions: 1) the nature of the research objective (exploratory, descriptive, or analytic); 2) the time frame under investigation (retrospective, cross-sectional, or prospective); and 3) whether the investigator intervenes in the events under study (observational or interventional). Therefore, we prefer to maintain current study designation.
9. Line 82: remove comma after the word “surveys”
Thank you form the suggestion. Changes were made according to recommendation.
10. Line 83: the last sentence here should go into the data analysis section.
Thank you form the suggestion. Changes were made according to recommendation.
11. Line 85: is the ethics committee the same as an IRB (Institutional Review Board)? If so, maybe give a short explanation. Also, specify that you obtained permission prior to any data collection.
We appreciate the comment. In our school we have an institutional review board (that approved the study) as well as an ethics committee that is responsible for the revision of all research proposals to ensure they agree with local and international ethical guidelines. Our study was also approved by this committee. At the end of the article there is an Institutional Review Board Statement where this is specified. Concerning permission to data collection we have revised the text to address your concerns and hope that it is now clearer.
12. For the paragraph starting line 84: explain more about how students gave consent. Were they given the option to participate or not? Was this study part of a course, or something completely separate from a course. How did you prevent coercion?
We appreciate the comment. To prevent coercion survey was disseminated by email and participants were free to participate / not to participate. This was an independent study not integrated in any course or curricular unit. Because we had a small sample, we were careful with sociodemographic data. For example, regarding age, participants had classes of age so no one could be identified (please see table 2). We have revised the text to address these issues.
13. Paragraph starting line 104: this paragraph could be more clear and concise.
Thank you for the suggestion. Changes were made according to recommendation.
14. Line 111: what was the cronbach’s alpha in the published study?
We appreciate the comment. In Lloyd et al study four factors were extracted that had high reliabilities (.892; .806; .805; .870). We have revised the text to address this issue.
15. For paragraph starting line 130: are these Cronbach alphas from the current study?
Thank you form the comment. Cronbach values are referred to the current study and were extracted from table 1. We have resided the text to make this information clearer.
16. For the paragraph starting line 139: more detail needs to be written that explains more about the recruitment and consent process, especially since students were used as subjects. Also, did you include only fully completed surveys? Or did you include partially completed ones as well?
Thank you for the comment. We have addressed this issue in section 2.1 Study design. Surveys were filled online and they were all fully completed. We have revised the text also in this section to address your concerns and hope that it is now clearer.
17. For line 153: I’m not clear on the results here; did the students and teachers complete the same surveys? If so, was it from the perspective of participation in DET? Not about usage of the faculty and participation for the students? This needs to be clarified.
Thank you for the comment. The same survey was used by both students and teachers. Dimensions evaluated were the same. The way of asking the questions was adapted to the user (student or teacher). There was also the concern to adapt the questions that intended to collect sociodemographic characteristics as explained in section 2.3 Instrument.
18. Line 207: I’m not convinced that the aim of the paper is to contribute to the improvement of educational processes….it seems the purpose is more likely aimed at gathering preliminary evidence related to the use of DET…
Thank you for the comment. Our main goal is to contribute to the improvement of educational process. To achieve it we have to start by understanding how DET is used in our context. This is why we defined as objective to evaluate the use of DET in non-presential class-room teaching, benefits, constraints, and implications for teaching-learning process in nursing students. Text was revised accordingly.
19. For paragraph starting line 207: I feel like your explanation related to why faculty aren’t using DET more frequently should be backed with evidence from the published literature. Is this related to their age, generation, etc.?
Thank you for the comment. We have revised the text to address this issue.
20. For paragraph starting 288: add more detail about how we can overcome constraints
We appreciate the comment and additional text was added to address this issue.
21. Line 329: need to include study design, convenience sample as limitations.
Thank you for the suggestion. We have revised the text to include this issue.
22. There should be citations in the conclusion section.
Thank you for the suggestion. We have revised the text to include this issue.
Reviewer 2 Report
The purpose of this study is to contribute to the improvement of the teaching-learning process in the nursing career through the use of TED in non-face-to-face teaching in the classroom. The objective is to evaluate the use of TED in non-face-to-face teaching in the classroom, the benefits, limitations and implications for the teaching-learning process in nursing students. The following questions were defined: how is the use of TED addressed in non-face-to-face teaching in the nursing classroom? What are the benefits and limitations of using DET? What are the implications for the teaching-learning process of nursing students?
Although the subject is attractive and relevant in current scenarios of university education, there are two methodological problems that are important to me when publishing in a quartile 1 journal like this one:
1. the sample is small, it is not homogeneous and although it performs statistical tests it is not comparable or extrapolable to compare teachers with students when the sample is 140 students and 23 teachers.
2. The data collection instrument they present is not validated, the only thing is a panel of experts, of 17 people that is not enough even if cronhbach's alpha is less than 1.
For all this and despite the fact that the wording is good and coherent, I do not think it can be published in this journal, taking into account that the discourse they elaborate during the discussion and the conclusions is oriented to extrapolation, something not compatible with the method used.
Author Response
Response to Reviewers
We would like to thank the editor and the reviewers for careful and thorough reading of this manuscript and for the thoughtful comments and constructive suggestions, which were considered in the new manuscript version.
After receiving the reviewer’s comments, changes were made in the manuscript as detailed below:
Reviewer 2
The purpose of this study is to contribute to the improvement of the teaching-learning process in the nursing career through the use of TED in non-face-to-face teaching in the classroom. The objective is to evaluate the use of TED in non-face-to-face teaching in the classroom, the benefits, limitations and implications for the teaching-learning process in nursing students. The following questions were defined: how is the use of TED addressed in non-face-to-face teaching in the nursing classroom? What are the benefits and limitations of using DET? What are the implications for the teaching-learning process of nursing students?
Although the subject is attractive and relevant in current scenarios of university education, there are two methodological problems that are important to me when publishing in a quartile 1 journal like this one:
- the sample is small, it is not homogeneous and although it performs statistical tests it is not comparable or extrapolable to compare teachers with students when the sample is 140 students and 23 teachers.
We thank you for the time spent assessing our work.
The objective of this work is to evaluate the use of DET in the dimensions previously mentioned. For the comparison of small samples it is recommended to use non parametric tests. Therefore, we used Phi/Cramer's V precisely because it is a test that is not affected by sample size. Our goal was not to extrapolate results, as this is an exploratory study and we agree that sample size does not allow it to be done. Manuscript was fully revised to assure we do not intend generalization.
2. The data collection instrument they present is not validated, the only thing is a panel of experts, of 17 people that is not enough even if cronhbach's alpha is less than 1.
Data collection instrument was based on previously published literature. To assure instrument validity prior to results interpretation we accessed content validity as explained on section 2.3. For reliability assessment Cronbach alpha coefficient was calculated as explained at the end of section 2.3. Values obtained are considered satisfactory according to the literature (Polit & Beck, 2017) particularly when considering that this was the first attempt to use the instrument. These procedures allow us to assure instrument validity.
For all this and despite the fact that the wording is good and coherent, I do not think it can be published in this journal, taking into account that the discourse they elaborate during the discussion and the conclusions is oriented to extrapolation, something not compatible with the method used.
We appreciate the time and effort spent assessing our work. Being an exploratory study, it does not intend to generalize results. This research allowed us to gather preliminary and relevant information on the use of DET, that will further contribute to the improvement of the teaching learning process in nursing students. It is also an original study as no previous research was found on this topic, in this population, making it suitable for a key one journal and of high interest to the scientific community. We have fully revised the manuscript and removed or rephrased all sentences that could indicate extrapolation. We hope this manuscript version fits to the JPM requirements and reviewers comments.
Round 2
Reviewer 1 Report
Thank you for addressing comments.
Author Response
We appreciate the careful and thorough reading of the second version of the manuscript.